# Comparison of Maternal Functioning between Iranian Mothers with and without Depressive Symptoms: A Case-Control Study

**DOI:** 10.3390/ijerph17103350

**Published:** 2020-05-12

**Authors:** Fatemeh Karami Chamgurdani, Jennifer L. Barkin, Carolann L. Curry, Mojgan Mirghafourvand

**Affiliations:** 1Department of Midwifery, School of Nursing and Midwifery, Tabriz University of Medical Sciences, Tabriz 977-5138947, Iran; fatemeh.karami1995@gmail.com; 2Department of Community Medicine, Mercer University School of Medicine, Macon, GA 31207, USA; barkin_jl@mercer.edu; 3Skelton Medical Library, Mercer University School of Medicine, Macon, GA 31207, USA; curry_cl@mercer.edu; 4Social Determinants of Health Research Centre, Faculty of Nursing and Midwifery, Tabriz University of Medical Sciences, Tabriz 977-5138947, Iran

**Keywords:** maternal functioning, postpartum, depression

## Abstract

Postpartum depression (PPD) has adverse effects on the mother’s ability to work, her relationships, performance in other roles, and caregiving ability. The aim of this study was to compare levels of maternal functioning between mothers with depression symptoms and those without. The participants of this case-control study included 80 postpartum women (*n* = 40 with depressive symptoms and *n* = 40 with no depressive symptoms) referred to health centers in Tabriz, Iran, in 2018–2019. The case and control groups were matched for the number of deliveries (first or second deliveries) and the type of delivery (vaginal or cesarean section). A sociodemographic questionnaire, Edinburgh Postpartum Depression Scale (EPDS), and Barkin Index of Maternal Functioning (BIMF) were used for data collection. The relationship between maternal functioning and depression was assessed by conducting independent *t-*tests and Pearson correlation tests in bivariate analysis and applying the general linear model (GLM) in a multivariate analysis. There was no statistically significant difference between the two groups in terms of sociodemographic information. The mean (SD) total scores of maternal functioning in the case (depressed) and control (non-depressed) groups were 63.4 (12.2) and 93.3 (13.0), respectively. According to the independent *t-*test, this value in the control group was significantly higher than that of the case group (mean difference: −30.0; 95% confidence interval: −35.6 to −24.3; *p* < 0.001). In terms of the domains of the BIMF, based on the independent *t-*test, the mean score of all domains in the control group was significantly higher than that of the case group, except for the mother-child interaction dimension (*p* = 0.219). Based on the Pearson correlation test, there was significant negative correlation between the PPD score and total score of BIMF (r = −0.79, *p* < 0.001) and its domains (r = −0.81 to −0.54, *p* < 0.001). In addition, based on the GLM with adjusting the sociodemographic and obstetrics characteristics, the score of maternal functioning was significantly lower in the case group compared to the control group [β = −30.1; 95% CI: −36.8 to −23.4; *p* = 0.001]. The results of this study indicate that women with depressive symptoms have lower maternal functioning. Depressed women may find it difficult to perform their maternal duties and take care of themselves and other family members due to the burden of the depressive symptoms. Therefore, early diagnosis and treatment of postpartum depression can play an important role in improving daily maternal functioning.

## 1. Introduction

During pregnancy and postpartum, changes in physical, emotional, and social dimensions occur. Adaptation in postpartum is a complex process and often requires reprioritization on the part of the mother and family members in order to accommodate and care for the newborn [1]. Postpartum depression (PPD) is one of the most common behavioral problems and mental disorders in women of childbearing age [2]. Common symptoms of PPD include depressed mood, mood swings, mild euphoria, irritability, fatigue, confusion, and crying [3,4]. PPD can disrupt the mother-infant relationship, impede the infant’s emotional and cognitive development [5], and delay the child’s long-term mental development [6]. PPD also has adverse effects on a mother’s ability to work, her relationships, her performance in other life roles, and her ability to take care of the baby [7,8]. In global statistics, 10% to 15% of new mothers develop PPD, which can range from being relatively mild to very severe [9]. The DSM-5 describes the onset of PPD as being between 4 and 6 weeks postpartum; however, in practice, clinicians will classify a positive depression screen at any point in the postpartum as PPD [10]. The prevalence of postpartum depression has been reported between 0.5% to 60.8% worldwide and 3.5% to 63.3% in Asian countries [11]. There are several factors that directly and indirectly contribute to PPD. Infant feeding type (breast feeding) and infant weight in the first 4 weeks are directly related to postpartum depression, while fetus number, preterm labor, initiation of lactation, maternal age, and birth weight are indirectly related to postpartum depression. Factors such as the number of deliveries, frequent cell phone exposure during pregnancy, and gestational hypertension disorders have both direct and indirect impacts on postpartum depression [2].

Depression status is not the only indicator of overall health and performance in the maternal role; it is also important to understand how postpartum women function in daily life and with added responsibilities. Optimal postpartum maternal functioning is possible when (1) the mother has adequate social support (social support), (2) is able to maintain her own physical (self-care) and mental health (psychological well-being), (3) is bonding with and caring for her infant (mother-child interaction, infant care), (4) can juggle her various responsibilities (management), and (5) adjusts appropriately over time (adjustment) [12]. Women often present for treatment with the goal of improved functioning rather than the achievement of a specific score on a depression scale [13]. Therefore, alternatives and compliments to depression assessment and treatment are required in order to accommodate women interested in a different approach to mental wellness [14]. Skill-building approaches that target improvements in daily functioning have been proven effective in new mothers and may also mitigate the effects of depression [15]. Treatments that positively impact both depression and functioning should be the gold standard. 

Functional assessment may represent a valuable addition to postpartum depression screening, providing a more thorough characterization of the mother’s health and quality of life [16]. While some literature elucidates the nature of the relationship between maternal functioning (measured via the Barkin Index of Maternal Functioning (BIMF)) and postpartum depression [16,17,18], little is known about this relationship in the Iranian new mother population. Even less is known about how the six functional domains defined by Barkin et al. (2010) [13] are affected by the presence of depression. Due to the established and deleterious impact of PPD on short- and long-term child development and the likely negative effects of impaired postpartum functioning, it is necessary to examine these variables in tandem. The aim of this study was to compare total maternal functioning scores and scores across the six functional domains in women who are depressed versus non-depressed. 

## 2. Material and Methods

### 2.1. Participants and Study Design

This case-control study was conducted with 80 postpartum women who were referred to health centers in Tabriz, Iran, from February 2018 to May 2018. Inclusion criteria were having a score of 13 or higher (case group) and a score of 12 and lower (control group) on the Edinburgh Postnatal Depression Scale (EPDS) [19], having ability to care for one’s infant, having a healthy term infant, having an infant with no neonatal abnormalities (whether diagnosed physical and mental abnormalities), and possessing the literacy level required to complete the questionnaires. Exclusion criteria were having cardiovascular disease, hypertension, liver disease, and other chronic diseases, or hospitalization of the child.

### 2.2. Sample Size Calculation

The sample size was calculated using G-power software and was based on the results of Posmontier et al.’s study in which maternal functional levels were compared in depressed versus non-depressed women [20]. The Inventory of Functional Status After Childbirth (IFSAC)—a precursor to the BIMF—was used to capture functional status in this study and resulted in a mean of 3.31 (m_1_) for non-depressed mothers and a mean of 2.76 (m_2_) for depressed mothers. The other inputs for the sample size calculation were the standard deviations, desired power, and alpha, specifically, sd1 = 0.73, sd2 = 0.83, α = 0.05 and power = 90%, which were used to calculate a sample size of 36 for both the non-depressed and depressed group of women. Factoring in a potential 10% attrition rate, the final sample size for both cases and controls was calculated to be 40, for a total of 80 women.

### 2.3. Sampling

After obtaining approval from the Ethics Committee of Tabriz University of Medical Sciences (ethics code: IR.TBZMED.REC.1398.218), convenience sampling was conducted in health centers (12 out of 80 health centers) with different socioeconomic characteristics. In order to represent different socioeconomic conditions, these 12 health centers were located in low-, middle- and high-income areas (four centers from each area). The researcher first selected mothers three weeks after childbirth who had recently had their first or second delivery based on their health record. After explaining the research and its importance through a telephone call, the researcher invited the prospective participants to the relevant health center on a specified date. The researcher retained those who were still interested in participating and these women were provided with more detailed information regarding study goals, logistics, and benefits of participation. Informed consent and the EPDS were completed for eligible women. Participants were then divided into two groups of mothers with depressive symptoms (case group) and mothers without depressive symptoms (control group) based on their EPDS scores. An EPDS score of 13 or more indicated the presence of depressive symptoms. The case and control groups were matched on the number of deliveries (first and second) and by type of delivery for the youngest child (vaginal or cesarean section). A sociodemographic and obstetrics questionnaire and the Barkin Index of Maternal Functioning (BIMF) [14] were also completed by the participants. A total of 160 mothers with first or second delivery based on their health record were assessed based on the eligibility criteria. Thirty-two mothers were excluded from the study due to being illiterate or having an elementary education level (n = 8), having chronic disease (n = 5), having preterm infants and their hospitalization history (n = 11), and unwillingness to participate in the study (n = 8), and 128 mothers completed the EPDS and were categorized into two groups based on the EPDS score (40 mothers with a score of 13 and higher were selected as case group and among 88 mothers with a score of 12 and lower, 40 mothers were selected as control group after matching with case group based on the number of deliveries and by type of delivery (Figure 1)).

### 2.4. Data Collection Tools

#### 2.4.1. Sociodemographic and Obstetrics Characteristics Questionnaire

The questionnaire included questions about infant gender, age of parents, education level of parents, employment status of parents, income adequacy, spouse and family support, and pregnancy intention (desired or unwanted). 

#### 2.4.2. Barkin Index of Maternal Functioning (BIMF)

The BIMF was used to assess postpartum maternal functioning. The self-report questionnaire was developed by Barkin et al. and has 20 items and 7 domains. The domains include self-care (items 2, 11, 13), infant care (items 12, 14), mother and child interaction (items 4, 5, 15), maternal psychological well-being (items 1, 2, 3, 5, 7, 10, 11, 16, 18, 20), social support (items 6, 8, 9), management (items 7, 11, 13, 14, 17, 18), and adjustment (items 17, 19) [12,13,21,22]. Each item is scored from 0 to 6 and the total score range is 0 to 120, with 120 representing optimal functioning. Though this questionnaire is intended to measure maternal functioning during the first postpartum year, it would likely be suitable for mothers of children up to 18 months old. The important features of the questionnaire are as follows: (1) it covers all areas related to postpartum functioning, (2) it is a patient-centered questionnaire (3), it has favorable psychometric properties, and (4) it is applicable in both clinical and research settings [14]. The psychometrics properties of this questionnaire have been confirmed in Iran [23]. In the present study, the Cronbach’s alpha coefficient was 0.88 and the intra-class correlation coefficient (ICC) was 0.85.

#### 2.4.3. Edinburgh Postnatal Depression Scale (EPDS)

The EPDS is the most widely validated tool for measurement of postpartum depression [24]. The questionnaire consists of 10 questions that are scored from 0 to 3; and the total score ranges from 0 to 30, with higher scores indicating greater levels of depressive symptomology. A total of 13 has been used as the clinical threshold in other Iranian studies and indicates a positive screen for depression [19,25]. The Iranian version of this questionnaire is an acceptable and reliable tool for measuring postpartum depression [19]. Montazeri et al. (2007) [19] determined the reliability of the questionnaire as 0.77 using Cronbach’s alpha method and as 0.8 using the test-retest method.

### 2.5. Analysis of the Data

Data was analyzed using SPSS version 24 software after being collected from all research units. The normality of quantitative variables was assessed using the K-S test; all were normally distributed. The independent *t-*test, chi-square, chi-square for trend, and Fisher exact tests were used to assess the homogeneity of the two groups in terms of sociodemographic characteristics. The relationship between maternal functioning and depression was assessed by conducting independent *t-*tests and Pearson correlation tests in bivariate analysis and the general linear model (GLM) in a multivariate analysis; *p* < 0.05 was considered statistically significant.

## 3. Results

A total of 80 women (40 with and 40 without depressive symptoms) were included in the study. The mean (SD) age was 29.1 (6.0) in the case group and 28.0 (5.4) years in the control group (Table 1).

The mean (SD) age of spouse was 33.2 (4.4) in the case group and 32.6 (4.8) years in the control group. The sex of the children in half of the cases was female (50.0%) and male (50.0%) in the case group. The majority of women in both groups (70.0% in case group and 77.5% in control group) had planned pregnancies. Half of the women in the case group (50.0%) and about half in the control group (45.5%) had a high school diploma. The majority of participants in both groups (92.5% in case group and 92.5% in control group) were homemakers. About one-third of spouses in both groups (35.0% in case group and 35.0% in control group) had freelance jobs. The majority of women in both groups (75.0% in case group and 67.5% in control group) reported that their monthly income was “somewhat enough” for living expenses (the sufficiency of monthly income was assessed through one subjective qualitative question that had three options to respond including “Completely sufficient”, “Somewhat sufficient”, and “Insufficient”). About one-third of the participants in both groups (27.5% in case group and 35.0% control group) had support from their spouses and families (32.5% in case group and 35.0% in control group) (Table 1).

The mean (SD) total scores of BIMF in the case and control groups were 63.4 (12.2) and 93.3 (13.0), respectively, which indicates significantly higher maternal functioning in the non-depressed group (control group) relative to the depressed group (case group) (mean difference: −30.0; 95% confidence interval: −24.3 to −35.6; *p* < 0.001) (Table 2).

In terms of the domains of the BIMF, based on independent *t-*tests the mean score of all domains in the control group was significantly higher than the case group. The only exception being the mother-infant interaction dimension, in which there was no statistical difference between the two groups (*p* = 0.219) (Table 2).

Based on the Pearson correlation test, there was significant negative correlation between PPD score and total score of BIMF (r = −0.79, *p* < 0.001) and its domains (r = −0.81 to −0.54, *p* < 0.001) (Table 3). In addition, based on the GLM with adjusting the sociodemographic and obstetrics characteristics, the score of maternal functioning was significantly lower in the case group compared to the control group [β = −30.1; 95% CI: −36.8 to −23.4; *p* = 0.001], and depression alone predicted 56% of the variance in the maternal functioning score. There was no relationship between sociodemographic and obstetrics characteristics with maternal functioning (Table 4).

## 4. Discussion

The results of this study demonstrated that the total, mean maternal functioning score in the group of mothers without depressive symptoms was significantly higher than the mean for the group of mothers with depressive symptoms. The magnitude of the difference between the two groups was also impressive as non-depressed women had BIMF scores that were on average 30 points higher. Also notable was that six of the seven BIMF functional domain scores were also significantly higher, on average, in the non-depressed group; mother-child interaction was the only domain where the scores were not significantly different between the two groups. This result, regarding maternal-child interaction, should be further explored.

In a cross-sectional study, results demonstrated a significant negative correlation between PPD and social activities and total score of maternal functional status and self-care [26]. The results of another study showed a 12-fold decrease in functional status of mothers with postpartum depression relative to their non-depressed counterparts [20]. Both aforementioned studies used the Inventory of Functional Status after Childbirth (IFSAC), which is mechanistically different than the tool used in our study. The BIMF includes self-care, infant care, mother-child interaction, psychological well-being, social support, management, and adjustment sub-domains, while the subdomains of the IFSAC are self-care, infant care, social activities, household chores, and professional activities. Self-care and social support scores were also more favorable among the non-depressed participants, on average, in the present study. This result is both intuitive and corroborative of the literature in which social support is well-established as being directly related to mental health [27]. These results indicate that measures to evaluate maternal functioning should be taken in mothers with PPD. In a study based on factors related to postpartum maternal functioning in depressed women, Barkin et al. (2016) found that health care providers dealing with the treatment of postpartum women with mental health problems (e.g., PPD) should be aware of the potential consequences to maternal functioning. Their results revealed the potential impact of maternal dysfunction on child development [16].

The results of these studies indicate that effective depression treatment might also positively impact maternal functioning levels [28]. In a clinical trial, Logsdon et al. (2003) showed that after 8 weeks of antidepressant treatment in mothers with postpartum depression, symptoms of depression and functional status were improved [7]. Logsdon et al. (2011) reached a similar conclusion in another prospective longitudinal observational study. They found that treatment of PPD improves maternal functioning [28].

In this study, there was significant negative correlation between PPD and maternal functioning. As mentioned above, a similar result has been reported in other studies [20,26]. A depressive state can cause an individual’s lack of interest in everyday activities and a lack of energy to manage tasks and do routine activities. Furthermore, depression may influence the ability of women to maintain relationships with family members and friends as well as their children [29]. Thus, depressive symptoms can lead to a decrease in maternal functioning.

Due to the importance of maternal functioning and its correlation with PPD, health care providers should be well-informed in these subject areas to improve maternal functioning in mothers through appropriate methods such as health education strategies [30] or counseling with the skills training approach [15]. In addition, conducting further quantitative and qualitative research regarding the inhibitors and facilitators of good maternal functioning is recommended to provide mothers with proper strategies. 

The sampling scheme for this investigation represents both a strength and a limitation. While the recruitment of women across a diverse set of health care settings represents a strength, the invocation of convenience sampling is not ideal. Another limitation is that we used psychological measures but not biological measures such as hair cortisol levels for measuring PPD. The relatively small sample size was another limitation of the present study. In addition, we did not assess the birth weight of infants as an effective factor on the study variables. Moreover, we used the standard and valid questionnaires of the BIMF (to assess postpartum maternal functioning) and EPDS. The usage of the BIMF and the EPDS is a strong point as both have excellent psychometric properties and have been validated in various populations. As this study included predominantly urban-dwelling women, future studies should include investigation of these same variables in rural populations. Finally, conducting studies with a large sample size is recommended. 

## 5. Conclusions

The present findings further highlight the importance of screening for depression, as it appears to correlate with daily, postpartum functioning and most domains of functioning. It is also possible that by addressing functional deficits first, through therapeutic skill-building interventions, depressive symptoms may lift as a result [15]. Because this study is cross-sectional, it is not possible to ascertain if the presence of depression is driving functioning scores or vice versa. Longitudinal studies may assist in establishing cause and effect. Due to the relationship between depression and maternal functioning, assessment of maternal functioning by healthcare providers may be a less stigmatizing way to screen postpartum women for depressive symptoms as soon as possible and a new strategy for effective interventions. Additional interventions that target improved daily functioning should be made available alongside pharmacotherapy or other traditional therapies for PPD.

## Figures and Tables

**Figure 1 ijerph-17-03350-f001:**
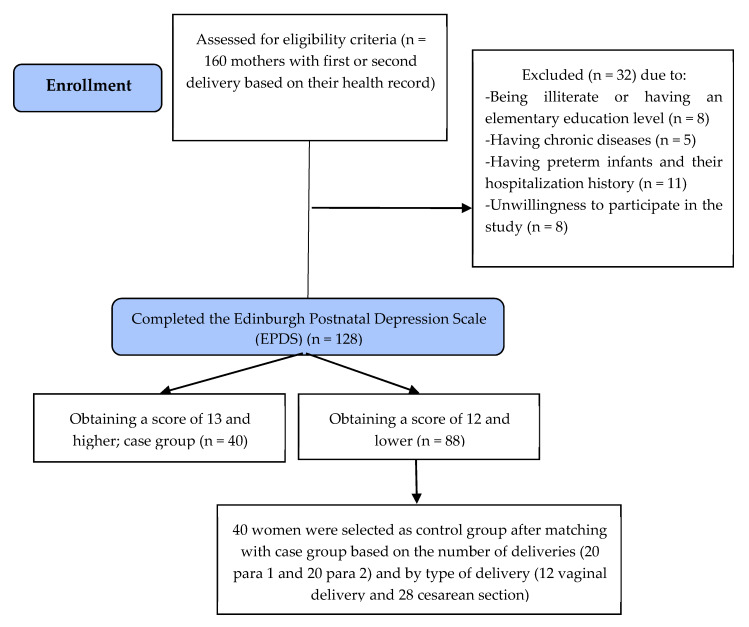
Flowchart of the study.

**Table 1 ijerph-17-03350-t001:** Sociodemographic and obstetrics characteristics in the study groups.

Variable	Case (*n* = 40)	Control (*n* = 40)	*p*-Value
	Number (Percent)	Number (Percent)	
Age ^#^	29.1 (6.0)	28.0 (5.4)	0.395 *
Husband age ^#^	33.2 (4.4)	32.6 (4.8)	0.562 *
Child sex			0.260 ^†^
Girl	20 (50.0)	15 (37.5)	
Boy	20 (50.0)	25 (62.5)	
Job			1.000 ^‡^
Housewife	37 (92.5)	37 (92.5)	
Employed	3 (7.5)	3 (7.5)	
Husband’s job			0.916 ^†^
Worker	12 (30.0%)	10 (25.0)	
Employee	9 (22.5)	9 (22.5)	
Shopkeeper	5 (12.5)	7 (17.5)	
Others	14 (35.0)	14 (35.0)	
Education			0.284 ^§^
Secondary school	10 (25.0)	6 (15.0)	
High school	2 (5.0)	4 (10.0)	
Diploma	20 (50.0)	18(45.0)	
Academic	8 (20.0)	12 (30.0)	
Husband’s education			0.365 ^§^
Elementary	5 (12.5)	1 (2.5)	
Secondary school	11 (27.5)	11 (27.5)	
High school	3 (7.5)	5 (12.5)	
Diploma	10 (25.0)	11 (27.5)	
Academic	11 (27.5)	12 (30.0)	
Sufficiency of monthly income for living expenses			0.060 ^§^
Completely sufficient	4 (10.0)	10 (25.0)	
Somewhat sufficient	30 (75.0)	27 (67.5)	
Insufficient	6 (15.0)	3 (7.5)	
Husband’s support			0.085 ^§^
Too much	11 (27.5)	14 (35.0)	
Much	9 (22.5)	14 (35.0)	
Medium	7 (17.5)	7 (17.5)	
Low	9 (22.5)	2 (5.0)	
Very little	4 (10.0)	3 (7.5)	
Family’s support			0.340 ^§^
Too much	13 (32.5)	14 (35.0)	
Much	8 (20.0)	11 (27.5)	
Medium	6 (15.0)	7 (17.5)	
Low	7 (17.5)	4 (10.0)	
Very little	6 (15.0)	4 (10.0)	
Unwanted pregnancy			0.446 ^†^
Yes	12 (30.0)	9 (22.5)	
No	28 (70.0)	31 (77.5)	

^#^ Mean (standard deviation). ^§^ Chi chi-trend test. * Independent test. ^‡^ Fisher’s exact test. ^†^ Chi-square test.

**Table 2 ijerph-17-03350-t002:** Comparison of maternal functioning and its dimensions in the case and control groups.

Variable	Case (*n* = 40) Mean (SD)	Control (*n* = 40) Mean (SD)	Mean Difference (95% Confidence Interval)	*p*-Value *
Total score of Maternal Functioning (Scale range: 0 to 120)	63.4 (12.2)	93.3 (13.0)	−30.0 (−35.6 to −24.3)	<0.001
Self-care (Score range: 0 to 18)	10.1 (2.8)	14.3 (3.6)	−3.7 (−5.6 to −2.7)	<0.001
Infant care (Score range: 0 to 12)	7.8 (1.7)	10.8 (2.4)	−2.8 (−3.7 to −1.9)	<0.001
Mother-child interaction (Score range: 0 to 18)	7.0 (1.6)	6.5 (1.6)	0.5 (−0.3 to 1.2)	0.219
Psychological well-being (Score range: 0 to 60)	29.8 (7.1)	42.8 (4.4)	−12.9 (−15.5 to −10.3)	<0.001
Social support (Score range: 0 to 18)	7.9 (3.7)	14.5 (3.6)	−6.6 (−8.2 to −4.9)	<0.001
Management (Score range: 0 to 36)	18.4 (4.4)	25.4 (4.7)	−7.0 (−9.0 to −5.0)	<0.001
Adjustment (Score range: 0 to 12)	7.5 (1.9)	10.6 (2.4)	−3.1 (−4.1 to −2.1)	<0.001

* Independent *t-*test.

**Table 3 ijerph-17-03350-t003:** Correlation of maternal functioning and its dimensions scores with postpartum depressive symptoms score.

Variable	Correlation with Postpartum Depressive Symptoms Score r (*p*-Value)
Total score of Maternal Functioning	−0.79 (<0.001)
Self-care	−0.62 (<0.001)
Infant care	−0.54 (<0.001)
Mother-child interaction	−0.81 (<0.001)
Psychological well-being	−0.74 (<0.001)
Social support	−0.71 (<0.001)
Management	−0.62 (<0.001)
Adjustment	−0.57 (<0.001)

**Table 4 ijerph-17-03350-t004:** The relationship between maternal functioning and depression based on general linear model (*n* = 80).

Variable	B (Confidence Interval 95%)	*p*-Value
Groups (Reference: Control)		
Case	−30.1 (−36.8 to −23.4)	<0.001
Age ^#^	−0.0 (−1.0 to 0.9)	0.976
Husband age ^#^	−0.4 (−1.5 to 0.8)	0.542
Child sex (Reference: Boy)		
Girl	2.0 (−5.9 to 10.0)	0.615
Job (Reference: Employed)		
Housewife	−10.0 (−24.3 to 4.4)	0.169
Husband’s job (Reference: Others)		
Worker	−1.1 (−10.2 to 8.0)	0.806
Employee	−3.6 (−16.2 to 8.9)	0.566
Shopkeeper	−5.1 (−15.6 to 5.3)	0.331
Education (Reference: Academic)		
Secondary school	−1.2 (−14.5 to 12.0)	0.850
High school	−7.2 (−23.1 to 8.8)	0.371
Diploma	3.6 (−6.4 to 13.7)	0.471
Husband’s education (Reference: Academic)		
Elementary	3.4 (−15.0 to 21.7)	0.714
Secondary school	8.0 (−6.4 to 22.4)	0.271
High school	1.3 (−14.2 to 16.7)	0.869
Diploma	4.2 (−8.5 to 16.9)	0.508
Sufficiency of monthly income for living expenses (Reference: Insufficient)		
Completely sufficient	5.2 (−8.7 to 19.1)	0.456
Somewhat sufficient	2.6 (−9.4 to 14.6)	0.666
Husband’s support (Reference: Very little)		
Too much	−1.6 (−22.3 to 19.2)	0.880
Much	−8.8 (−27.5 to 9.8)	0.346
Medium	−11.5 (−29.0 to 6.1)	0.195
Low	−6.0 (−24.9 to 13.0)	0.530
Family’s support (Reference: Very little)		
Too much	−0.6 (−20.0 to 18.7)	0.947
Much	9.0 (−9.5 to 27.5)	0.333
Medium	7.2 (−10.8 to 25.2)	0.427
Low	4.3 (−11.8 to 20.3)	0.596
Unwanted pregnancy (Reference: No)		
Yes	3.9 (−5.0 to 12.7)	0.383

^#^ Adjusted R^2^ = 56%.

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
