# Peer review of "Comparison of Maternal Functioning between Iranian Mothers with and without Depressive Symptoms: A Case-Control Study"

_ijerph, 2020, doi:10.3390/ijerph17103350_

Round 1
Reviewer 1 Report
Would like to see what the implications of the findings are discussed in the conclusion. Make the methods a clearer
Author Response
Response to reviewers
We thank you for their highly insightful comments which have enabled us to greatly improve the quality of our manuscript. In the following are our point-by-point responses to each of the comments.
Reviewer 1
- Would like to see what the implications of the findings are discussed in the conclusion. Make the methods a clearer.
Thank you for your time investment in our work. We have written the implications of the findings at the end of Conclusion section and also fifth paragraph of Page 7. Also, we have read the methods again and it was clear.
Thanks

Reviewer 2 Report
Journal: IJERPH
Title: Comparison of Maternal Functioning between Iranian Mothers with and without Depressive Symptoms: A Case-control Study
Manuscript Number: IJERPH-779775
Keywords: maternal functioning; postpartum; depression
Thank you for the opportunity of reading and reviewing this interesting study.
Maternal Functioning betweenIranian Mothers with and without DepressiveSymptoms.
The study is well written. I only have some major and minor issues below.
The authors sometimes use double references: For example in line 195 : "In a cross-sectional study, results from Fathi et al. (2015) [25]". Please use only the Vancouver system throughout the text.
Abstract:
Line 17: Please re-write to "n = 40 with depressive symptoms and n = 40 with no depressive symptoms"
Introduction:
The introduction is very well written.
Line 79: I would say "The aim of this study was [...]!. The study has been already done, so it´s better to use the past tense.
Methods:
Why did you use as exclusion criteria having cardiovascular disease, hypertension, liver disease, and other chronic diseases, or hospitalization of the child?
Line 93: Please provide a reference for Posmontier et al.’s study.
Taking into consideration that postpartum depression happens in 10-15 women out of 100, to get a sample of 40 women with postpartum depression symptoms means that at least 400 women were approached. Why did the rest (at least 360 women) of those women did not want to participate in the study? You could have used those potential participants to compare if their sociodemographic data differed from those agreeing to participate.
Results:
The analyses strategy seems relatively simple. I would try adding for example a lineal regression, including a figure in the manuscript, to see if any of sociodemographic information could predict postpartum depression symptoms or maternal functioning in both groups. Besides, adding a Pearson correlation table will add some additional information to this study.
Discussion:
The discussion is well-written, but it is quite short. Please add at least a couple of paragraphs explaining potential implications of this study in clinical settings and research.
Also include in the discussion the results obtained in the previously suggested statistical analyses.
Further limitations of this study should be included. For example, only used psychological measures but not biological measures (hair cortisol levels is a promising biomarker previously used when exploring postpartum depression in different countries).
Author Response
Reviewer 2
Thank you for the opportunity of reading and reviewing this interesting study. Maternal Functioning between Iranian Mothers with and without Depressive Symptoms. The study is well written. I only have some major and minor issues below.
We appreciate the dear reviewer.
- The authors sometimes use double references: For example in line 195 : "In a cross-sectional study, results from Fathi et al. (2015) [25]". Please use only the Vancouver system throughout the text.
We have used the Vancouver system throughout the text.
Abstract:
- Line 17: Please re-write to "n = 40 with depressive symptoms and n = 40 with no depressive symptoms"
We have re-written it.
Introduction:
The introduction is very well written.
Line 79: I would say "The aim of this study was [...]!. The study has been already done, so it´s better to use the past tense.
Thank you for your time investment in our work. We have used the past tense.
Methods:
- Why did you use as exclusion criteria having cardiovascular disease, hypertension, liver disease, and other chronic diseases, or hospitalization of the child?
Because cardiovascular disease, hypertension, liver disease, and other chronic diseases, or hospitalization of the child can influence on maternal functioning, therefore, we have considered them as exclusion criteria.
- Line 93: Please provide a reference for Posmontier et al.’s study.
We had cited a reference for Posmontier et al.’s study previously (Reference number 20). We have highlighted it.
- Taking into consideration that postpartum depression happens in 10-15 women out of 100, to get a sample of 40 women with postpartum depression symptoms means that at least 400 women were approached. Why did the rest (at least 360 women) of those women did not want to participate in the study? You could have used those potential participants to compare if their sociodemographic data differed from those agreeing to participate.
We appreciate your observations in this regard. However, as we had determined sample size for both cases and controls equal with 40. Therefore, we have included 40 participants with no depressive symptoms in our study and after inclusion of controls, the sampling continued until the 40 cases (women with depressive symptoms) participated in the study. Of course, based on a systematic review and meta-analysis, the pooled prevalence of PPD was %28.7 and in another study that conducted in 2018, the prevalence of PPD was 35.4% in Iran.
Veisani, Yousef and Sayehmiri, Kourosh. Prevalence of Postpartum Depression in Iran - A Systematic Review and Meta-Analysis. The Iranian Journal of Obstetrics, Gynecology and Infertility, 2012, 15 (14), 21-29.
Ezzeddin N, Jahanihashemi H, Zavoshy R, Noroozi M. The Prevalence of Postpartum Depression and Its Association with Food Insecurity among Mothers Referring to Community Health Centers. Iran J Psychiatry. 2018;13(4):280-287.
Results:
The analyses strategy seems relatively simple. I would try adding for example a lineal regression, including a figure in the manuscript, to see if any of sociodemographic information could predict postpartum depression symptoms or maternal functioning in both groups. Besides, adding a Pearson correlation table will add some additional information to this study.
Since we have compared the sociodemgraphic variables between groups, and there was no statistically significant difference between the two groups in terms of sociodemographic information. Therefore, we didn’t done linear regression for adjusting the sociodemographic information. However, we have assessed the correlation between depression and maternal functioning by Pearson correlation test and reported its results in the Results section and Table 3.
Discussion:
The discussion is well-written, but it is quite short. Please add at least a couple of paragraphs explaining potential implications of this study in clinical settings and research. Also include in the discussion the results obtained in the previously suggested statistical analyses.
We have explained potential implications of this study in clinical settings and research. Also, we have included the results obtained based on the Pearson correlation test in the discussion section.
- Further limitations of this study should be included. For example, only used psychological measures but not biological measures (hair cortisol levels is a promising biomarker previously used when exploring postpartum depression in different countries).
We have included the mentioned limitation.
Thanks

Reviewer 3 Report
Manuscript ID: ijerph-779775
Type of manuscript: Article
Title: Comparison of maternal functioning between Iranian mothers with
and without depressive symptoms: a case-control study
This is an interesting study. The manuscript is well written.
The aim of the study is to compare levels of maternal functioning between mother with depression symptoms and those without. They reported lower maternal functioning with depression symptoms > or =13.
The number of women included in the study is relatively small.
The authors chose the EPDS with a cut-off of 13 for depressive symptoms. They classified women into two groups: depressed and control. It seems difficult to choose such a cut-off to determine women with depression symptoms and controls. Did the authors look for a correlation between EPDS scores and BIMF scores? It could be interesting to present those results.
Several data could influence maternal functioning. For example, do the authors have data about the baby.
In the discussion, a limitation section should be added.
Author Response
Reviewer 3
- This is an interesting study. The manuscript is well written. The aim of the study is to compare levels of maternal functioning between mother with depression symptoms and those without. They reported lower maternal functioning with depression symptoms > or =13.
We appreciate the dear reviewer.
- The number of women included in the study is relatively small.
We agree with dear reviewer. We have mentioned it as a limitation of the present study.
- The authors chose the EPDS with a cut-off of 13 for depressive symptoms. They classified women into two groups: depressed and control. It seems difficult to choose such a cut-off to determine women with depression symptoms and controls. Did the authors look for a correlation between EPDS scores and BIMF scores? It could be interesting to present those results.
Based the following references, an EPDS score of 13 has been used as the clinical threshold in Iranian studies for indicating a positive screen for depression. We have assessed the correlation between depression and maternal functioning by Pearson correlation test and reported its results in the Results section and Table 3.
Montazeri, A.; Torkan, B.; Omidvari, S. The Edinburgh Postnatal Depression Scale (EPDS): translation and validation study of the Iranian version. BMC Psychiatry 2007, 7, 11.
Kheirabadi, G.R.; Maracy, M.R.; Akbaripour, S.; Masaeli, N. Psychometric properties and diagnostic accuracy of the edinburgh postnatal depression scale in a sample of Iranian women. Iran J Med Sci 2012, 37, 32-38.
- Several data could influence maternal functioning. For example, do the authors have data about the baby.
The case and control groups were matched on the number of deliveries (first and second) and by type of delivery for the youngest child (vaginal or cesarean section). There was no statistically significant difference between the two groups in terms of sociodemographic information that reported in Table 1. We have only assessed the child sex and there was no significant difference in terms of this variable between groups.
- In the discussion, a limitation section should be added.
The last paragraph of Discussion section is about strengths and limitations of study.
Thanks

Round 2
Reviewer 2 Report
Thank for letting me reviewing this new version of the manuscript.
Although it has been improved, the authors failed to answer some of the issues the last version had.
My main concerns are:
- How did the authors collected the group of postpartum depression group: Although the authors report a high prevalence of postpartum depression in their country, they are not clear in this point. I strongly recommend to provide a detailed description of sample collection and using a flow diagram to clarify this issue.
- Authors have added a Pearson correlation, which is a good thing. But analyses are still quite simple. Please, as recommended before, I recommend them to do more sophisticated analyses.
Author Response
- How did the authors collected the group of postpartum depression group: Although the authors report a high prevalence of postpartum depression in their country, they are not clear in this point. I strongly recommend to provide a detailed description of sample collection and using a flow diagram to clarify this issue.
Thank you. We have provided a flow diagram to clarify this issue. Details regarding recruitment/selection of the participants can be found on lines 123-130
Authors have added a Pearson correlation, which is a good thing. But analyses are still quite simple. Please, as recommended before, I recommend them to do more sophisticated analyses.
We have used adjusted General Linear Model and its results were reported in the results section of abstract and main text as well as Table 4. The analytic techniques that we employed were appropriate for addressing the primary study aims.
Thanks

Reviewer 3 Report
The authors improved their manuscript.
However, I think the most interesting result is the negative correlation. The authors should added them in the abstract section.
Several data could influence maternal functioning. For example, do the authors have data about the baby. I was thinking about prematurity, low birth weight and neonatal hospitalisation. This should be added in the limitation section.
I think it is important to screen not only major depression but also depressive symptoms as soon as possible. So, I am embarrassed by the new proposal in the last sentence. I will delete it. I think one of the core message could be to search for depressive symptoms as soon as possible.
line 184: please correct BIMF instead of BIME.
Author Response
The authors improved their manuscript.
However, I think the most interesting result is the negative correlation. The authors should added them in the abstract section.
We have added the results about the negative correlation in the abstract section.
Several data could influence maternal functioning. For example, do the authors have data about the baby. I was thinking about prematurity, low birth weight and neonatal hospitalisation. This should be added in the limitation section.
Having healthy term infant was an inclusion criteria and also neonatal hospitalization was an exclusion criteria. As showed in the flow diagram, five mothers were excluded due to having exclusion criteria. However, we didn’t assess the birth weight of infants and reported it as a limitation.
- I think it is important to screen not only major depression but also depressive symptoms as soon as possible. So, I am embarrassed by the new proposal in the last sentence. I will delete it. I think one of the core message could be to search for depressive symptoms as soon as possible.
We have revised it.
- Line 184: please correct BIMF instead of BIME.
We have corrected it.
Thanks

Round 3
Reviewer 2 Report
Thank you for the opportunity of reviewing this review version of the manuscript.
I thank the authors for their effort to improve their manuscript.
All the suggestions have been adequately addressed.
I only have one minor issue. Please add in the discussion the findings related to the regression analyses performed.